# The Changes in Red Blood Cell Indices That Occur in Pre-Diabetic Patients of all Ethnicities from the 25–45 Years of Age: A Protocol for a Systematic Review and Meta-Analysis

**DOI:** 10.3390/mps6010013

**Published:** 2023-01-24

**Authors:** Nomusa Christina Mzimela, Aubrey Mbulelo Sosibo, Phikelelani Siphosethu Ngubane, Andile Khathi

**Affiliations:** 1School of Laboratory Medicine and Medical Science, College of Health Sciences, University of Kwa-Zulu Natal, Durban 4000, South Africa; 2Nomusa Mzimela; Department of Human Physiology, School of Laboratory Medicine and Medical Sciences, College of Health Sciences, University of KwaZulu-Natal, Private Bag X54001, Durban 4000, South Africa

**Keywords:** systematic review, meta-analysis, pre-diabetes, red blood cells, inflammatory markers, hematologic changes

## Abstract

**Introduction**: Pre-diabetes is an intermediate, asymptomatic state between normoglycaemia and the onset of type 2 diabetes mellitus (T2D). Recent reports indicate that there are sub-clinical changes observed in red blood cells during pre-diabetes. This systematic review protocol will provide an outline of all procedures in the synthesis of the available data on the changes in red blood cell indices. **Methods and Analysis:** This protocol was prepared by adhering to the PRISMA 2015 guidelines for reporting protocols. Published clinical studies that involve observation, whether it is cross-sectional, comparative cross-sectional, case-control or cohort study designs that involve normal/non-diabetic and pre-diabetes reports were used. Additionally, this was accomplished by using clinical MeSH headings to search on MEDLINE, COCHRANE library and African Journal Online. Three reviewers (NCM, AMS & AK) screened all the results for eligibility criteria. Then, Downs and Black checklist was used to check the risk of bias. Review Manager v5.4 Forrest plot was used for meta-analysis and sensitivity analysis. Strength of evidence was then assessed using the Grading of Recommendations Assessment, Development, and Evaluation approach (GRADE). **Results and Conclusion:** This protocol will give direction on the exploration of articles that report on changes in red blood cell indices in the pre-diabetic state. The results obtained from this protocol will further give direction on the research to be done at in the eThekwini district of South Africa. **Ethics and Dissemination:** The data that will be analyzed will be data that has already been published thus there will be no data collection from subjects. Therefore, no ethical clearance is required. **Registration Details**: This protocol has been registered with the International Prospective Registry of Systematic Reviews (PROSPERO) registration number “**CRD42020189080**” dated 05-07-2020.

## 1. Background

Pre-diabetes is an intermediate, asymptomatic state of moderate insulin resistance that often occurs before the onset of type 2 diabetes mellitus T2D [1]. It is characterised by fasting blood glucose (FBG) from 5.6 to 7.0 mmol/L; 2 h postprandial blood glucose (2 h—OGTT) from 7.8 to 11.0 mmol/L and glycated haemoglobin (HbA1c) levels from 5.7% to 6.4% [2,3]. According to the International Diabetes Federation (IDF) atlas reports, T2D accounts for approximately 90% of all diabetes cases globally, thus highlighting the need for further studies to better understand its pathology [4]. The onset of T2D is, however, often preceded by pre-diabetes [5]. IDF statistics reported that there were 19 million diabetic people in Africa aged between 20 and 79 years in 2019 [4]. Surprisingly, according to IDF reports, 12 million Africans aged between 20–79 years were reported to live with undiagnosed T2D in 2019 [4]. The age range with the highest diagnosis of T2D in South Africa is 45–65 years while pre-diabetes is said to last anywhere between 10 and 20 years [6]. Therefore, in order to investigate changes in the pre-diabetic state, we chose to look at patients in the age range of 25–45 years old. One of the complications of T2D is a reduction in red blood cell (RBC) deformability and changes in concentration of RBC contributing to changes in blood indices [7,8,9]. Reports indicated that T2D patients display changes in RBC indices such as mean corpuscular volume (MCV), mean corpuscular haemoglobin (MCH), mean corpuscular haemoglobin content (MCHC), haemoglobin (HGB) and haematocrit (HCT) [9,10,11]. Additionally, Nada reported that people with T2D also have impaired erythropoiesis which is indicated by low levels of erythropoietin (EPO) [10]. The RBCs in T2D have also been shown to have reduced deformability and a reduced lifespan [9,12]. Studies further indicate that red blood cell distribution width (RDW) in T2D is increased due to anisocytosis and RBC degradation [10]. Moreover, increased T2D RBCs aggregation is also reported to cause an increase in blood viscosity and the development of high blood pressure [13]. This then contributes to the development of cardiovascular complications due to the clogging of vessels [14,15,16]. Furthermore, according to Sharif et al., anemia is the key indicator of chronic kidney disease, cardiovascular disease and retinopathy [17]. It is currently a debatable issue if these complications occur during pre-diabetes. In addition to the available data, studies from our laboratory showed changes in RBC indices in the pre-diabetic state using a diet-induced pre-diabetes animal model [18,19,20,21,22]. This research from our laboratory raised questions as to whether the same abnormalities could be observed in human subjects considering the limitations in the diet-induced pre-diabetes animal model. From the search carried out, we could not obtain any report or evidence of a systematic review that reports on the changes in red blood cell indices and the level of secretion of EPO and endothelial nitric oxide synthase (eNOS) in the pre-diabetic state in the eThekwini district, Durban, South Africa. Therefore, this presents an opportunity to deliver a systematic review that will yield a comprehensive synthesis obtained from the available collected studies that previously reported on the red blood cell indices and concentration of EPO and eNOS during pre-diabetes.

### Objectives

To determine the changes in RBC indices’ (RBC, MCV, MCH, MCHC, HGB, HCT and RDW) concentration in the pre-diabetic state.To investigate if there are changes in concentration with respect to WBC, EPO and eNOS in the pre-diabetes state.To determine the impact of demographics on hematologic changes and secretion of EPO and eNOS in the pre-diabetic state.

## 2. Methods

This protocol was prepared by adhering to the preferred reporting items for systemic reviews and meta-analysis (PRISMA) 2015 guidelines for reporting protocols (PRISMA checklist attached in additional file).

### 2.1. Systematic Review Registration

The protocol has been registered with the International Prospective Registry of Systematic Reviews (PROSPERO registration number “CRD42020189080” dated 05-07-2020).

### 2.2. Eligibility Criteria for the Study

Studies with a minimum of 100 study participants that report community-based clinical cross-sectional study will be eligible. The inclusion and exclusion criteria will be as follows.

*Inclusion*: Information that is obtained from non-diabetic adults within the ages of 25–45 of all ethnicities will be eligible.

*Exclusion*: The study will not use reports from people with a history of any blood disease, liver disease, kidney disease, heart disease or depression. Additionally, reports from pregnant women will also not be used. Articles from professional sports athletes will not be allowed in the study.

### 2.3. Pre-Diabetes Diagnosis Criteria

The diagnostic criteria that will be used are in line with the criteria used by the American Diabetes Association [3]. Pre-diabetes diagnostic criteria will be as follows (participants used in reports should meet one of the following diagnoses): fasting blood glucose (FBG): 5.6–7.0 mmol/L; 2 h postprandial blood glucose (2 h—OGTT): 7.8–11.0 mmol/L with Glycated haemoglobin (HbA1c): 5.7–6.4%.

### 2.4. Study Design

#### Information Sources

*Participants*: The target of the information source will be any reported clinical study that involves a minimum of 100 study participants, of all genders, aged from 25 to 45 years from all ethnicities. The number of 100 as a minimum is recommended in order to increase statistical power and reliability.

*Intervention*: The clinical studies that involve observational studies if they will be cross-sectional, comparative cross-sectional, case-control, or cohort study designs that involve non-diabetic and pre-diabetes reports. The reported information that involves specifically one or more RBCs indices (RBC, MCV, MCH, MCHC, HGB, HCT and RDW) at the pre-diabetic stage will be eligible for this systematic review. Additionally, studies that report information that involves WBC, EPO and eNOS will also be eligible for this systematic review.

*Comparators*: In this systematic review, the eligible comparing control groups will be non-diabetic/normal control.

## 3. Outcomes

This systematic review is expected to have outcomes as follows. The primary outcomes:The changes in RBC indices’ (RBC, MCV, MCH, MCHC, HGB, HCT and RDW) concentration in the pre-diabetic state (reported as odds ratios and 95% confidence interval).The changes in concentration of WBC, EPO and eNOS in the pre-diabetic state (reported as odd ratios and 95% confidence interval).

The secondary outcome:3.Changes in RBC indices, EPO and eNOS markers due to demographic impact such as the effect of gender, age, and race (reported as the mean).

### 3.1. Search Strategy

To identify studies involving cohorts, the electronic search strategy will be used that is related to the study of interest. This strategy will be accomplished by searching on MEDLINE (from 1963 to 2020), COCHRANE library displaying results of trials from PubMed, CT.gov, EMBASE, and ICTRP (from 1963 to 2020), and African Journal Online (from 1998 to 2020). Additionally, to these search strategies, the use of clinical MeSH headings and text words will be applied to filter the available information. For all searches done, the keywords to be used will be “pre-diabetes and erythrocytes,” “pre-diabetes and red blood cells,” “pre-diabetes and red blood cell indices,” “pre-diabetes and red blood cell parameters,” “pre-diabetes and erythropoietin” and “pre-diabetes and endothelial nitric oxide synthase.”

### 3.2. Identification of Eligible Studies

NCM, AMS & AK will then screen the title and abstracts of all the obtained results, and the studies that meet the eligibility criteria will then be selected. Each reviewer will be responsible for screening all the selected study reports before decision-making regarding eligible reports. The PRISMA flowchart for the selection of studies will then be provided on reports from the systematic review.

### 3.3. Patient and Public Involvement

No patients are involved.

### 3.4. Data Management

#### 3.4.1. Study Records and Data Extraction

A Microsoft Excel file will be used to record the extracted data of study records selected as eligible reports. The pre-defined list of variables to be considered in each report will be used as categories in the Excel file. Considering the research of interest, the outcome of interest will mainly be the RBC indices’ response and concentration of EPO and eNOS in both genders, at an age parameter of interest in all ethnicities. Additionally, the value of the baseline characteristic of the data reported will also be considered. Therefore, the baseline characteristics of eligible research reports obtained will be author, year of publication, country, and study setting. The methodology of the study reported will also be considered with the categories (design, period, sampling strategy, and whether participants are normal or pre-diabetic population) considered. Finally, the outcomes from different gender, ages, ethnicities, RBC indices changes/markers will then be extracted.

#### 3.4.2. Data Simplification

For the simplification of data, the studies that report on the RBC indices (RBC, MCV, MCH, MCHC, HGB, HCT, and RDW) will be grouped into a single group. Additionally, the studies that report on WBC, EPO and eNOS will also be grouped into a single group.

### 3.5. Risk of Bias

To measure the potential risk of bias in individual studies, the Downs and Black checklist will be used [23]. For clarity, the scores will be rated as follows: excellent (25), good (20–24), moderate (14–19), poor (11–13) and very poor (<10). Three reviewers (NCM, AMS and AK) will be responsible for the independent judgments which will be based on the four domains of the Black and Downs checklist tool which is reporting bias (10 items), external validity (3 items), internal validity (6 items), and selection bias (7 items). In a situation where there will be a difference in opinions between NCM, AMS and AK, PSN will then be responsible for adjudication.

### 3.6. Data Synthesis

For the meta-analysis of reported data, a Review Manager version 5.4 software Forrest plot will be used [24,25]. Using this RevMan forest plot, eligible data from all reported studies will be meta-analyzed depending on the sample size and the odds ratio of the RBC indices (RBC, MCV, MCH, MCHC, HGB, HCT and RDW) or markers (EPO and eNOS) and WBC in both pre-diabetic and control groups. Additionally, an odd ratio and confidence interval will be used to plot the forest plot where the solid lines will represent the 95% confidence interval. Each reported study will be represented as a horizontal line on the *y*-axis to list the primary author and year of study. The forest plot will also include the weight of the study results that will be automatically obtained using RevMan software.

### 3.7. Sensitivity Analysis

Heterogeneity will also be automatically calculated using the RevMan software forest plot. The greater homogeneity will be indicated by a greater overlap between the confidence intervals [25]. Using the forest plot, I^2^ will be calculated where a value between 0% and 100% will be obtained. Additionally, a value obtained less than 25% will be an indication of a strong homogeneity, and a value obtained greater than 75% will then be an indication of a strong heterogeneity. However, a value of 50% will be considered as an average value.

### 3.8. Assessment of Strength of Evidence

Assessment of the strength of evidence will be done by NCM, AMS and AK. The studies included in the review will then be evaluated using the Grading of Recommendations Assessment, Development, and Evaluation approach (GRADE) [25,26,27]. Furthermore, using a GRADE pro tool, a summary of the findings (SoF) table will then be created.

## 4. Discussion

This protocol is compiled to give a direction to the systematic review and meta-analysis. In this systematic review and meta-analysis, the focus will be on the globally published reports that will be based on the RBC indices and selective markers in the pre-diabetic state. However, to be able to provide a complete systematic review, it is of great interest to compile a protocol that will provide the methods to be used during the analysis of reports to be collected. The synthesis of previous study reports obtained from this systematic review and meta-analysis will provide clarity of the contribution of RBC indices (RBC, MCV, MCH, MCHC, HGB, HCT and RDW) on hematologic changes during pre-diabetes. This systematic review and meta-analysis will also give a synthesis of data from previous reports based on markers, that are EPO and eNOS and the WBC concentration. Additionally, the synthesis from this systematic review and meta-analysis will analyze any existing correlations with demographics on hematologic changes in the pre-diabetic state.

## Data Availability

No extra data is available besides the attached additional file since it is a protocol for systematic review.

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
