# Peer review of "The Changes in Red Blood Cell Indices That Occur in Pre-Diabetic Patients of all Ethnicities from the 25–45 Years of Age: A Protocol for a Systematic Review and Meta-Analysis"

_mps, 2023, doi:10.3390/mps6010013_

Round 1

Reviewer 1 Report

The paper tried to review changes in red blood cell indices that occur in prediabetic patients. This is an interesting topic and has clinical significance. However, the paper did not describe the result of the literature review.    

Reviewer 2 Report

This study evaluated The changes in red blood cells indices that occur on pre-diabetic patients of all ethnicities, from the age of 25 to 45 years: A protocol for a systematic review and meta-analysis. Though the research title is interesting. There are several shortcomings that need to be addressed.  if the following points are corrected, it can be suitable for publication in the journal.

1- There are several English language grammatical and typographical mistakes that should be rectified.

2- The registered title in the PROSPERO is not the same as the article title. Please explain it.

Abstract

3- The result and conclusion section should be rewritten Because it is written in summaries.  

Methods

No comments.

Discussion

8- The discussion section is more summarize and needs to be improved

Round 2

Reviewer 1 Report

I have no additional comments on this revised paper.